# Application of a Natural Antioxidant from Grape Pomace Extract in the Development of Bioactive Jute Fibers for Food Packaging

**DOI:** 10.3390/antiox10020216

**Published:** 2021-02-02

**Authors:** Cristina Cejudo-Bastante, Paloma Arjona-Mudarra, María Teresa Fernández-Ponce, Lourdes Casas, Casimiro Mantell, Enrique J. Martínez de la Ossa, Clara Pereyra

**Affiliations:** Chemical Engineering and Food Technology Department, Wine and Agrifood Research Institute (IVAGRO), University of Cádiz, Puerto Real, 11519 Cádiz, Spain; cristina.cejudo@gm.uca.es (C.C.-B.); paloma.arjonamudarra@alum.uca.es (P.A.-M.); teresafernandez.ponce@gm.uca.es (M.T.F.-P.); lourdes.casas@uca.es (L.C.); enrique.martinezdelaossa@uca.es (E.J.M.d.l.O.); clara.pereyra@uca.es (C.P.)

**Keywords:** active packaging, food preservation, supercritical solvent impregnation, vinification by-products, natural fibers

## Abstract

There is an increasing demand for the use of new food packaging materials. In this study, natural jute fibers impregnated with a Petit Verdot Red Grape Pomace Extract (RGPE) was proposed as a new active food packaging material. Pressurized Liquid Extraction (PLE) and Enhanced Solvent Extraction (ESE) techniques were employed to obtain the bioactive RGPE. Afterward the supercritical solvent impregnation conditions to obtain RGPE-natural jute fibers were studied, by varying pressure, modifier percentage and dried RGPE mass. PLE technique offered the highest bioactive extract at 20 MPa, 55 °C, 1 h residence time using C_2_H_5_OH:H_2_O (1:1 *v*/*v*), providing an EC50 of 3.35 ± 0.25 and antibacterial capacity against *Escherichia coli, Staphylococcus aureus* and *Pseudomonas aeruginosa* (MIC of 12.0, 1.5 and 4.0 mg/mL RGPE respectively). The natural jute fibers impregnated with 3 mL of that RGPE (90 mg/mL) at 50 MPa and 55 °C generated the most efficient packing material with regards to its food preservation potential.

## 1. Introduction

Grape cultivation is one of the main and most widespread agro-economic activities in the whole world, mostly used for winemaking, with an official global production in 2019 in excess of 77 million tons according to FAO statistics. Wine production generates enormous amounts of by-products. Its valorization is mainly represented by the production of soil fertilizers, fermentation substrate for biomass production and livestock feed [1]. However, there are certain restrictions on the reuse of these by-products. For example, the phytotoxicity of certain polyphenols could have an antimicrobial effect during composting, which would impair their use for such purpose. Regarding their use as cattle feed, some animals have shown intolerance to certain components, such as condensed tannins, which negatively affect their digestion [2]. However, grape vinification by-products have a high content in bioactive compounds, especially polyphenols and condensed tannins (proanthocyanidins) or even anthocyanins, in the case of red grape pomace [3], which represent a source of antioxidant and antimicrobial rich compounds of interest for the manufacturing of different cosmetic, pharmacological or food products [4,5]. For instance, it has been reported the use of aqueous extracts of red grape pomace in the elaboration of meat products due to their antioxidant and antimicrobial properties, particularly against mesophiles, psycrotrophics and fecal microbiota [6,7,8]. However, the applicability of RGPE go beyond their direct addition in food formulation and it has been also investigated in material engineering. Besides its use as a natural dyeing colorant for wood, silk, cotton, polyamide or acrylic fabrics [9,10,11], anthocyanins and proanthocyanidins from red grape and other sources have been recently used as pH-sensitive compounds in intelligent packaging and as an active substance in active film-packaging formulation [12,13,14,15]. For example, Kurek et al. evaluated the antioxidant properties of chitosan and carboxymethyl cellulose films enriched with blueberry and red grape pomace extracts [16]. Bi et al. completed a deep study and demonstrated not only the antioxidant properties but also the antimicrobial ones of a chitosan-proanthocyanidins film against *Escherichia coli, Salmonella spp., Staphylococcus aureus* and *Listeria monocytogenes* during in vitro experiments [17]. In agreement, Qin et al. discuss the same properties on biodegradable films enriched with purple corn extract [18] and *Lycium ruthenicum* Murr. [19], which demonstrated, in that case, anthocyanins’ preservative effect on pork meat. Moreover, Xu et al. evaluated the antimicrobial capacity against *S. aureus* and *L. monocytogenes* of the Cabernet Franc and Viognier grape pomace extracts used for the production of starch films [20].

Today’s society as well as some governmental regulations, point the absolute necessity to replace plastic materials by other biodegradable alternative constituents. In the food field, the highest challenge is the replacement of plastic-packaged and net-packaged products by other more environmentally friendly alternatives. In this sense, the functionalization of natural fibers by supercritical fluids seem a reliable alternative due to its resistant nature grants a lower modification of their structure under supercritical conditions. Among them, jute (*Corchorus capsularis*) is one of the most natural fibers used in food packaging, because it has been traditionally used to pack bulk vegetables and beans since decades. It has been used for different applications such as packaging paper, decorations, reinforcement for polymers, construction, furniture, automotive applications [21,22]. Although some research studies on fabrics, such as cotton, with natural extracts for biomedical application have been reported [23,24,25,26], as far as we know, there is scarce research on the use of jute in the food industry. The literature found on this subject is focused on the use of natural fibers as a reinforcing filler in composites [27,28], because their improvement of strength, stiffness and better moisture absorption [29]. Just to mention an example, Jawaid et al. [30] developed a material based on jute in combination with oil palm fruit bunches and obtained a resistant composite material that could be used for different applications. Moreover, Gangopadhyay et al. [31] developed a polypropylene (PP) and superabsorbent fiber (SAF) based on technical fabric, to be used as a potential packaging material to transport fresh fruit and vegetables. Furthermore, Chatterjee et al. (2020) developed a laminated packaging made of jute fibers and PP by thermal processing [32]. However, no active substances were included in any of those formulations.

In view of the literature and the great potential of both natural fibers and RGPE for food industry, this study intends to analyze the development of an alternative jute food-packaging material with bioactive properties thanks to its impregnation with RGPE. Among the different techniques reported in the bibliography, the supercritical solvent impregnation (SSI) is the one that is having a greater scientific impact at the moment. On one hand, scCO_2_ offer advantages in comparison to other traditional techniques such as casting or extrusion, like their applicability to different matrices (wounds, polymers, particles, etc.) or that being a functionalization technique after the polymerization of the matrix do not compromise the polymerization step [33]. Besides, it offers the possibility to combine both SFE and SSI in a continuous system [34,35]. Numerous investigations have focused on the supercritical impregnation of natural extracts, such as thyme [36], clove [37,38] or oregano essential oils [39] among others, into synthetic polymeric films.

In the present study, the RGPE is obtained by a Pressurized Liquid Extraction (PLE) and Enhanced Solvent Extraction (ESE) has been used as supercritical extraction techniques, due to they combine the use of polar co-solvents and scCO_2_ to increase the extraction yield of plant extracts [40,41].

The objective of the present study is both analyze the extraction and impregnation process using high pressure techniques. Extraction yield and antioxidant capacity levels of the RGPE obtained by ESE and PLE were compared by varying different pressure (10 and 20 MPa) temperature (55−70 °C) and co-solvent (C_2_H_5_OH or C_2_H_5_OH:H_2_O) conditions. The antimicrobial capacity of the chosen extract against food-borne pathogens was analyzed prior to its impregnation into jute fibers, in order to obtain active fabrics with potential preservative properties as food packaging. For this purpose, the influence of some supercritical impregnation parameters were studied, such as pressure (10−50 MPa) percentage of modifier (2.8, 4.8 and 6.7%) and amount of dried RGPE (28.8 and 270 mg) in the impregnation vessel.

## 2. Materials and Methods

### 2.1. Chemical Reagents, Raw Materials and Bacterial Samples

The chemical reagents and materials used in this work are presented in Table 1. Red grape pomace (RGP) of the Petit Verdot variety was supplied by “Bodegas Luis Pérez” (Jerez de la Frontera, Spain). It was obtained immediately after the vinification process and it was dried in an oven at 60 °C. Prior to the extraction process, it was grinded with a blender to reduce particle size. Finally, natural jute fibers with a grammage of 305 g/m^2^ was supplied by Deyute (Tejijut S.L.U, Crevillente (Alicante, Spain)).

### 2.2. High-Pressure Extraction of RGPE

The extractions were carried out in a high-pressure equipment (Thar Technologies SF100, Pittsburgh, PA, USA) described in a previous work [42] (Figure 1). 20 g of RGP was loaded into a paper filter cartridge and it was installed inside the extraction vessel. When the pre-set temperature conditions were achieved, the co-solvent and the CO_2_ supply were pumped until the desired pressure was reached. Then, the BPR opened to let the extract enter the cyclonic separator. In the PLE procedure, no CO_2_ process-line was necessary, since only liquid solvents are used.

The extraction conditions were selected according to the results obtained in previous studies [41]. First, an ESE process was completed during 1 h using CO_2_ and ethanol pumped at 5 g/min each one, with a total flow rate of 10 g/min. Pressure and temperature were studied at 10 and 20 MPa, at 55 and 70 °C respectively. PLE was carried out at the best pressure and temperature conditions previously defined by ESE, using pure ethanol and a mixture of C_2_H_5_OH:H_2_O (1:1 *v*/*v*) as solvents. The extraction yields were calculated (Equation (1)):(1)$$\mathrm{Extraction}\;\mathrm{yield}=\frac{m_{\mathrm{RPGE}}\;(g)}{m_{\mathrm{RGP}}\;(g)}\;\times \;100$$
where $m_{\mathrm{RGPE}}$ is the mass of the extract obtained and $m_{\mathrm{RGP}}$ is the mass of red grape pomace extracted.

### 2.3. Supercritical Impregnation of Natural Jute Fibers with RGPE

The impregnation process was carried out in the same high-pressure set-up described in Section 2.2, but using a thermostatic flat bottomed impregnation vessel (104 mL) (Waters Corp., Milford, MA, USA). A certain amount of RGPE was poured into the vessel. A magnetic stirrer, running at 60 rpm, is installed at the bottom of the vessel to aid to the dissolution of the extract. Then, it was introduced a stand that holds a 5 cm wide rounded-shape sample of jute natural fibers horizontally. The impregnation procedures were carried out in batch mode. CO_2_ was first pumped at 10 g/min until the desired pressure conditions were reached. Then, the CO_2_ flow was stopped and the system pressure was maintained until the impregnation time (1 h) was over. The system was rapidly depressurized (10 MPa/min) to obtain the impregnated jute fibers (IJF). The depressurization rate flow acted also as drying agent, so any further drying step of the fabric was required.

Three sets of experiments were carried out to determine the influence of some variables on the process (Table 2). The optimal impregnation conditions in each experiment were established according to the impregnation loading and antioxidant capacity of the impregnated natural fibers.

### 2.4. Analysis of the Bioactivity of the Extracts and the IJF Samples

#### 2.4.1. Antioxidant Capacity

The antioxidant capacity of samples was evaluated by means of a DPPH assay considering the methods described by Brand-Williams [43] and Scherer and Godoy [44]. The reaction of the N∙ radical of DPPH in the presence of the extract is controlled by reducing the absorbance at 515 nm. To perform the analysis, 0.1 mL of the extract at different concentrations was added to 3.9 mL of the 6 × 10^−5^ M DPPH ethanolic solution. The absorbance was measured every 2 min during 4 h until the plateau was reached. The tests were performed in triplicate.

The concentration of the remaining DPPH after the test was determined using a DPPH calibration line previously determined (Equations (2) and (3)):*Abs* = 10089*M* + 0.0085; R² = 0.9994(2)
(3)$$\mathrm{RemainingDPPH}=\frac{C_{\mathrm{DPPHf}}}{C_{\mathrm{DPPHi}}}\;\times \;100$$
where *M* is the molar concentration of DPPH, *C_DPPHi_* is the concentration of DPPH at the initial time and *C_DPPHt_* is the concentration of DPPH at time *t*. By plotting the values of the remaining %DPPH in stationary state versus the concentration of the extract at each point, the EC50 value (efficient concentration) can be graphically determined. The extract with the highest antioxidant properties was further characterized in terms of antibacterial activity and was used for the impregnation process.

The antioxidant capacity and loading of impregnated samples were determined using the method described by Cejudo et al. [33] with some changes. A specific amount of the impregnated natural fibers was introduced in 3 mL of ethanol and was sonicated during 30 min in order to extract the impregnated compounds. Then, the solvent was evaporated in a rotavapor and it was replaced by 4 mL of the DPPH reagent at 6 × 10^−5^ M in ethanol. The reaction was kept in the dark for 4 h and the reduction of the absorbance was determined as %I (Equation (5)). The results were expressed as %I/100 mg natural fibers. To calculate the antioxidant activity, the inhibition percentage (%I) (Equation (5)) versus the extract concentration at the pleateau was plotted, obtaining Equation (6):(4)$$I\;=\;\frac{{\mathrm{Abs}}_{i}{-\mathrm{Abs}}_{f}}{{\mathrm{Abs}}_{i}}\;\times \;100$$
*%I* = −0.3427*C*^2^+ 10.596*C* + 15.665; R² = 0.9989(5)
where *Abs_i_* is DPPH absorbance at the initial time, *Abs_t_* is DPPH absorbance at time *t* measured at 515 nm, and *C* is the RGPE concentration that results in a particular *%I* at each point.

In order to calculate the impregnation loading of the natural fibers samples, the concentration of RGPE was calculated expressing the results as µg RGPE/100 mg natural fibers.

#### 2.4.2. Antibacterial Capacity of the RGPE and the IJF Samples

The antibacterial capacity was tested against *E. coli*, *P. aeruginosa* and *S. aureus,* by agar dilution method in 60 mm Petri plates using 0.375, 0.75, 1.5, 3, 4, 5, 6, 8, 10 and 12 mg/mL of RGPE. The test was carried out in triplicate and a solution of DMSO and distilled water (1:1 *v*/*v*) was used as the negative control. The plates were inoculated with 0.1 mL of a 1.5 × 10^6^ UFC/mL inoculum. After a 24 h incubation at 37 °C, the minimum inhibitory concentration (MIC) in each bacterial culture was determined.

In the case of the impregnated samples, their inhibition properties were assessed in a liquid medium. Two different amounts of impregnated natural fibers (50 and 100 mg) were used in order to determine their inhibitory properties in two ranges. Samples were sterilized by UV and then introduced in a 5 mL sterilized tube with liquid LB propagation medium. 24 h were allowed for the compounds to diffuse in the culture media, and then the blank of each tube medium was measured at 625 nm. The extract compounds were considered to have diffused completely, achieving concentrations in the media of ca. 0.07 mg/mL and 0.14 mg/mL, when using 50 and 100 mg of IJF respectively. Then, the tubes were inoculated at 1.5 × 10^6^ UFC/mL and incubated (24 h/37 °C). Absorbance was measured at 625 nm to determine bacterial growth inhibition (Equation (6)).
(6)$$\mathrm{Inhibition}\;=\;\left(1\;-\;\frac{C_{f}}{C_{i}}\right)\;\times \;100$$
where *C_f_* is the cell concentration of samples and *C_i_* is the cell concentration in the positive control containing non-impregnated natural fibers.

### 2.5. Phenolic Characterization of RGPE and Impregnated Jute Fibers by UPLC-ESI-TOF-MS

The phenolic characterization of both the RGPE and the impregnated fibers have been carried out by ultra-performance liquid chromatography (UHPLC) coupled to quadrupole-time-of-flight mass spectrometry (QToF-MS) (Xevo G2 QToF, Waters Corp.). The chromatographic method followed have been previously described by Cejudo et al. (2018) [45], with some modifications. To recover phenolic compounds, certain amount of IJF was introduced into 10 mL of ethanol and sonicated during 30 min. Then, the fraction was evaporated in a rotavapor and diluted in 1 mL of ethanol and filtered through 0.22 µm prior to the chromatographic analysis. Not impregnated jute fibers have been also analysed in order to eliminate interferences in the phenolic determination, so the signals detected were used as blank of the impregnated jute fiber signals. Identification of compounds was carried out considering the molar mass, the molecular formula and the presence of compounds reported in literature [46,47]. Analysis have been done in duplicate.

The column employed in the analysis was an Acquity UPLCr BEH C18 column (50 mm × 2.1 mm i.d., 1.7 mm particle size, Waters Corporation). The mobile phase is comprised by a phase A (water with 0.1% formic acid) and a phase B (acetonitrile with 0.1% formic acid) working with a flow rate of 0.6 mL/min. Analysis was started at 98% phase A for 0.3 min, changing to 65% at 1.5 min later. Then solvent B achieved the 100% after 1.5 min, maintaining that proportion during 1 min. Then, the B percentage decreased until 2% and was maintained for a minute. The total method time was 5 min. Electrospray was operated in full scan analysis (100–1000 Da), working in negative ionization mode for the determination of phenolic acids, flavanols and flavonols, and in positive ionization mode for anthocyanin determination. Three calibration lines were carried out to quantify three different families of phenolic compounds: phenolic acids (Equation (7)), flavonols and flavanols (Equation (8)), and anthocyanins (Equation (9)):Gallic acid = 1813x − 1235.2; R² = 0.9997(7)
Quercetin = 221.68x − 737.86; R² = 0.9902 (8)
Cyanidin = 48.521x − 248.2; R² = 0.9951(9)

### 2.6. Scanning Electron Microscopy (SEM)

Natural fibers were visually evaluated by SEM (Quanta 200, FEI, Hillsboro, OR, USA) to verify the presence of the impregnated extract in the fibers. Samples analyzed were those obtained at the most convenient impregnation conditions. Fibers were coated with a gold layer (15 μm thick) and were submitted to a 20 kV voltage.

## 3. Results and Discussions

### 3.1. RGPE Bioactivity

The extraction yield and the antioxidant activity of the extracts obtained is collected in Figure 2. Respecting to the extraction yield of the extracts obtained by ESE, under isobaric conditions an increase in temperature from 55 °C to 70 °C hindered the extraction process. This behaviour can be attributed to the fact that the density of the mixture CO_2_+ C_2_H_5_OH decreased [48], which would intervene in the extraction of high-weight molecules as anthocyanins. Another possible explanation to this behaviour is the possibility of being in a retrograde solubility zone above this temperature. On the contrary, under isothermal conditions, an increase in pressure from 10 to 20 MPa improved extraction thanks to the higher density level of the mixture. Regarding to the extract bioactivity, the extract obtained at 20 MPa and 55 °C showed a significantly higher EC50 than the rest. At 10 MPa, the influence of the temperature was not as evident as the samples obtained at 20 MPa, where the increase of the temperature decreases the antioxidant capacity of the extracts. Mantell et al. [49] previously reported a higher extraction yield of anthocyanins from the same raw material in ESE when using methanol in higher % organic co-solvent instead of water, obtaining better recovering at 10 MPa rather than 50, and 60 °C rather than 40 °C. Thus, the employ of higher temperature studied in this case (70 °C) seemed to interfere the polyphenol recover. The presence of polar solvents was reported to have a positive effect for the extraction of compounds from different natural products [50,51]. In fact, organic solvents with water are usually combined in the isolation of bioactive compounds from grapes and their by-products [52,53]. For instance, Kytrité et al. [54] studied the recovery of antioxidant compounds of lingoberry by different SFE techniques (SFE-PLE), among which PLE-ethanol offered better recoveries of polar compounds above PLE-water. Accordingly, PLE was evaluated as a potential technique for the extraction of polyphenols and anthocyanins. The extractions took place at the most convenient pressure and temperature conditions previously determined by ESE (20 MPa and 55 °C) using different solvents. As can be seen (Figure 2), the use of C_2_H_5_OH:H_2_O would favour a slightly greater extraction yield. This result agrees with the research of Santos et al. when recovering active compounds from feijoa peel [55]. Moreover, the antioxidant capacity of the RGPE obtained by means of PLE increases significantly, providing very much higher antioxidant extracts, due to the EC50 values are quite low, leading to think that the mixture of organic solvent and water promoted the extraction of the active compounds. According to Mustafa and Turner [56], the ethanol in this dual mixture could improve the solubility of the analyte, while the water would enhance its desorption and would break the matrix–analyte hydrogen bonding. Moreover, the presence of water could be an important factor for the extraction of glycosylated anthocyanins ―as will be further discuss in Section 3.3.―, while the presence of ethanol decreases the water surface tension, favouring infiltration and mass transfer [56]. Corrales et al. [57] reported that the antioxidant capacity of RGPE obtained by PLE was specially enhanced by hydroalcoholic mixtures when ethanol content remained within the 50–80% range. Similar conclusions were reported by Otero-Pareja [41] obtaining better RGPE by PLE at 10 MPa and 120 °C. In agreement, Ferri et al. [58] reported a higher polyphenol extraction yield in RGPE of Merlot variety using PLE instead of solvent extraction using C_2_H_5_OH:H_2_O, although obtaining similar antioxidant activity. Alañon et al. studied the composition and bioactivity of wine by-products obtained by PLE, pointing the hydroalcoholic mixtures as better solvents for obtaining antioxidant and rich-in-phenolic compounds extracts [59]. Considering the results, the extract obtained by PLE at 20 MPa and 55 °C using C_2_H_5_OH:H_2_O was used in impregnation experiments, and was characterized in terms of antibacterial capacity.

Table 3 shows the Minimum Inhibitory Concentration (MIC) of RGPE against three common food pathogens. *Escherichia coli* was the most resistant bacteria to the presence of RGPE, with a MIC value of 12 mg/mL of RGPE, followed by *Pseudomonas aeruginosa* (4 mg/mL) and *Staphylococcus aureus* (1.5 mg/mL). Possibly, the double-lipid membrane of gram-negative bacteria makes them especially resistant to the action of some antibacterial substances [60]. The results obtained in this work are more satisfactory than other reported in literature. Respecting to the raw material, the grape variety influences the bioactivity of the extracts. Cheng et al. studied the antimicrobial activity of the grape pomace of Pinot noir and Pinot meunier [53], observing that *S. aureus* was also the most sensitive microorganism in all extracts studied, showing a MIC of 0.75 and 12.5 respectively in C_2_H_5_OH:H_2_O extracts. Regarding to the extraction technique used, lower activity was found in Petit Verdot extracts obtained from the maceration of seeds and grape skin, where no antibacterial activity against *E. coli* was found, and *S. aureus* was inhibited only when using concentrations over 6.25 mg/mL [61]. In fact, Oliveira et al. [62] found lower antimicrobial activity in Syrah and Merlot grape pomace extracts obtained by SFE at 25 MPa and 60 °C, observing no inhibition against *E. coli* and P. aeruginosa so, again, it seemed more convenient the use of PLE technique instead of other SFE technique. The literature comparing antibacterial activity of grape by-products by different conditions of PLE is very scarce, but there are references using other natural extracts. For instance, Zandoná et al. [63] studied the antimicrobial activity of araçazeiro (*Psidium cattleianum*) leaf by PLE (10.3 MPa, 50 °C) using different solvents, achieving higher inhibition of *S. aureus* and *P. aeruginosa* in water/ethanol mixtures rather than water or ethanol alone. All this information confirms the use of C_2_H_5_OH:H_2_O mixtures by PLE as the most convenient alternative for their further use in the impregnation experiments.

### 3.2. Bioactivity of RGPE-Impregnated Natural Fibers

The aim of the supercritical impregnation is to include an active substance into a polymeric matrix in order to obtain a new material with active properties. To evaluate the bioactivity of the jute fabrics impregnated with RGPE, the impregnation yield, the antioxidant activity and the antibacterial activity were determined.

Three different experiments were carried out in order to evaluate the influence of the operational variables in the impregnation process (Figure 3). The first experiment was carried out using 2.8% of modifier (C_2_H_5_OH:H_2_O (1:1)) and 28.8 mg of dried RGPE, to evaluate the influence of pressure variations on the process (Figure 3A). Greater pressure values provided impregnated natural fibers with higher RGPE loading and antioxidant capacity. When the pressure raises, the density of the supercritical fluid increases, which may favour both the swelling of the matrix and the solubility of the compounds. This behaviour agrees with the literature on the impregnation of polyester and cotton natural fibers with natural extracts, as well as on the impregnation of PET/PP films with natural extracts [42,45,64]. Based on these results, a pressure of 50 MPa was selected for the experiments to be carried out later on.

In the second experiment, the modifier percentage introduced in the impregnation vessel was evaluated at a fixed RGPE concentration (Figure 3B). This factor was studied at three particular levels (i.e., 2.8, 4.8 and 6.7%), while RGPE concentration remained constant (9.6 mg/mL). For this purpose, different amounts of dried RGPE were introduced in the vessel (28.8, 46.1 and 67.2 mg respectively). It was observed that an increase of the C_2_H_5_OH:H_2_O percentage had in general a negative effect on the impregnation loading while the best results were obtained with a modifier percentage of 2.8%, although no differences were found between using 2.8 and 4.8%. It has been reported that the relation CO_2_/modifier, as well as the nature of the modifier, interfere in the affinity between the solute and the matrix [64]. For instance, García-Casas and others established that 10% *v*/*v* of acetone/DMSO (80%/20%)/CO_2_ were the optimal conditions to impregnate mangiferin into silica [65]. The characteristics of the supercritical phase varied according to the percentage of modifier used. When increasing, the affinity between the solute and the supercritical phase seemed to increase, which eventually reduced the impregnation loading. Besides, the presence of water in the modifier could further evidence this change of affinity. The low solubility of water in scCO_2_ could lead to the formation of a second phase in the solvent [66], which may further alter the balance solute-fluid-matrix at higher modifier percentages. Regarding the effect of the modifier on the matrix, it has been reported that a higher modifier content increases the swelling effect of supercritical CO_2_ on polymers [67], which favours the impregnation of compounds. As far as we know, any literature of the swelling effect on jute natural fibers under supercritical conditions has been ever reported. Yet, such effect did not seem to be predominant in the impregnation efficiency. On the contrary, the amount of dried extract introduced in the vessel at a fixed modifier percentage showed a high relevance (Figure 3). When a greater amount of extract was used, the impregnation loading and the antioxidant capacity of the natural fibers increased considerably. Possibly, the supercritical phase was more concentrated in compounds and the system tended to balance by impregnating the matrix, producing natural fibers with a higher bioactivity. As far as we know, any similar results have been reported in natural fibers, although, some experiments carried out in polymers achieved the same conclusion. For instance, Wenzel et al. obtained better loading of walnut husk ethanolic extract in impregnated LDPE films when its concentration increased 1g/g ethanol [68]. Moreover, previous experiments with olive leaf extract impregnated into PET/PP polymer achieved the same conclusion [69], although the significance was not as evident as in this case, probably because of the high affinity between RGPE and the jute natural fibers.

Considering the results obtained, it was determined that the best impregnation conditions were 50 MPa and 55 °C, using a RGPE with 2.8% of C_2_H_5_OH:H_2_O and 270 mg of dried RGPE. The IJF used for the antibacterial assay was obtained at such optimal conditions.

Different pathogenic bacteria were cultivated in liquid LB medium in the presence of two amounts of impregnated natural fibers (50 and 100 mg) that resulted in a corresponding RPGE concentration in the culture media of 0.07 and 0.14 mg RGPE/mL respectively. This concentration of RGPE in the media was far below the MIC values reached by crude RGPE (Table 3) and too low to totally inhibit the growth of microorganisms. However, the natural fiber showed higher growth inhibition that would be expected, which might be due to a reduction in the RGPE particle size or by a selective impregnation of the RGPE compounds with greater antibacterial capacity. This behaviour had been previously observed when the antibacterial capacity of olive leaf extract against *P. aeruginosa* before and after its impregnation in PET/PP films [69].

### 3.3. Phenolic Composition of RGPE and Impregnated Jute Fiber

The quantification of the identified compounds is collected in Table 4. Eleven phenolic compounds were identified in the RGPE obtained by PLE using C_2_H_5_OH:H_2_O as extraction solvent. The abundance of flavonols is representative in the extract, which is in agreement to Jara-Palacios et al. [70], who found the quercetin-3-*O*-glucoside as the most abundant compound identified in a C_2_H_5_OH:H_2_O RGPE. On the other hand, it has been reported a great amount of catechin in GPE obtained by subcritical water extraction at 100 MPa at different temperature ranges [71]. In this case, although catequin is not the most abundant compound, it has demonstrated the highest affinity both with the supercritical phase and the fibers during impregnation, and it has been successfully loaded in the fibers.

From the RGPE compounds identified, only four have been found in the IJF, which evidence the selective character of the impregnation process, being especially selective to phenolic acids. Syringic, *p*-coumaric and protocatechuic acids were the most abundant compounds in impregnated fibers at the condition studied, being the responsible for their bioactivity. Regarding flavanols, only catechin was found in the impregnated fibers. The process was less selective to anthocyanins, and any of them was identified in the jute fabric. According to the results obtained by Buratto and researchers in the supercritical drying of aerogels impregnated with hydroalcoholic açai extract, polyphenols were lower affine to the supercritical phase than anthocyanins, promoting its impregnation. Therefore, anthocyanins were dragged out of the system and its remaining content in the dry aerogel was very low [72]. Possibly, this behavior also occurs in the jute impregnation, and the process was more selective in low-weight phenolic compounds, since significant content of catechin and *p*-coumaric acid were determined in IJF. Those compounds have been reported to have a high antioxidant and antimicrobial activity against food pathogens, such as *Staphylococcus aureus*, *E. coli* and *B.*
*cereus*, which confirms the results obtained in terms of the bioactivity observed in both the fabrics and the extracts [73,74].

### 3.4. SEM Images

Samples were evaluated by SEM in order to determine the distribution and the appearance of the extract in the natural fiber. SEM images at different magnifications are depicted in Figure 4.

The impregnated natural fiber (Figure 4C) showed a more rugged surface in comparison with the one of the untreated natural fiber (Figure 4A y Figure 4B), observing a partial penetration of the RGPE particles into the fibers and thus, the great affinity between RGPE and the fibers on the impregnation process. This result agrees with previous literature about the impregnation of mango leaf extract into natural cotton natural fibers, where spherical particles embedded into the fibers were observed [75].

## 4. Conclusions

The promising prospects in the production of active food packaging using agriculture by-products as active agents, opens a new way for the revalorization of such by-products. The RGP that is obtained after the vinification process still contains a large amount of bioactive compounds that can be put to some use after being extracted. Moreover, the solid residues after such extraction could be still used for other applications based on its protein, fiber or carbohydrate contents among others, where this target compound would be an intermediate product.

Regarding the production of RGPE, PLE using C_2_H_5_OH:H_2_O as the solvent seemed to be the best choice to obtain an extract with high antioxidant and antibacterial capacities. In view of the results, the antioxidant and antimicrobial properties of RGPE were transferred successfully to jute natural fabrics through supercritical impregnation. These results are comparable with other obtained in polymeric matrices, and offer a very encouraging view of the use of this natural fiber as active food packaging.

## Figures and Tables

**Figure 1 antioxidants-10-00216-f001:**
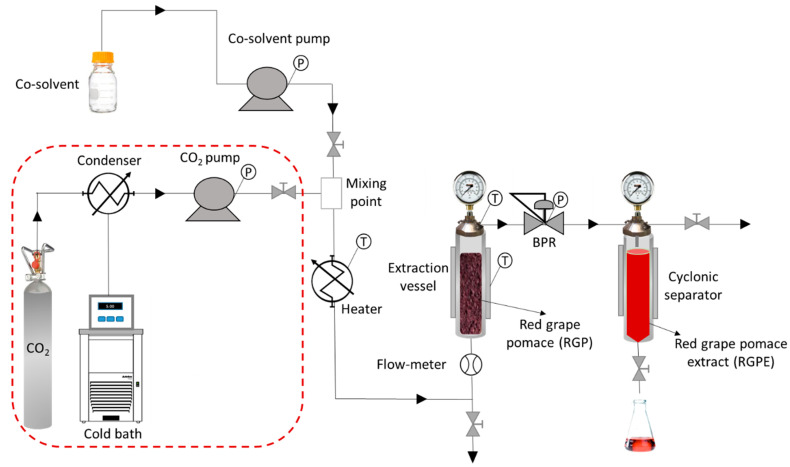
High Pressure Extraction equipment flowchart. The CO_2_ process-line (within the red dot-line) is not used in PLE processes.

**Figure 2 antioxidants-10-00216-f002:**
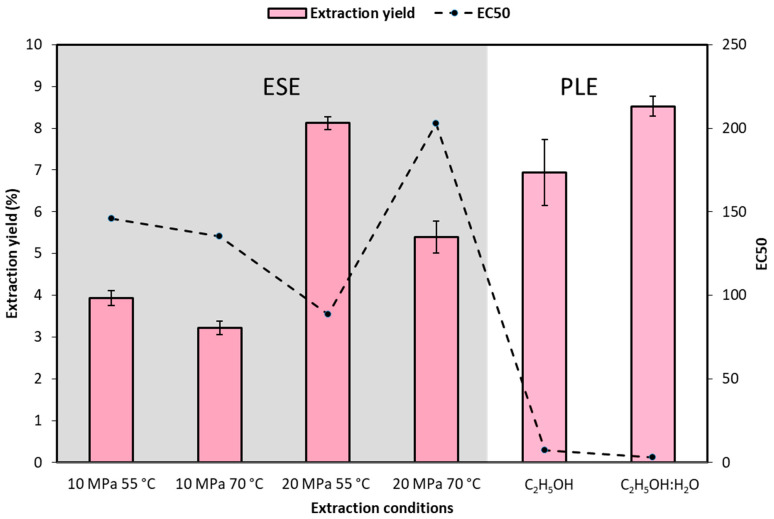
Extraction yield and antioxidant activity of the red grape pomace extracts (RGPE) obtained by ESE and PLE (*n* = 2).

**Figure 3 antioxidants-10-00216-f003:**
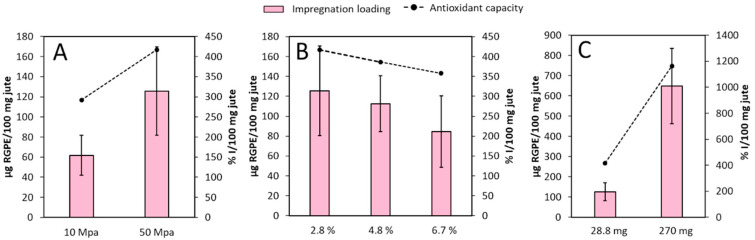
Influence of variables on the impregnation loading and the antioxidant capacity of the impregnated fabric samples (*n* = 2). (**A**) Effect of pressure (experiment 1). (**B**) Effect of modifier percentage (experiment 2). (**C**) Effect of dried RGPE loading (experiment 3).

**Figure 4 antioxidants-10-00216-f004:**
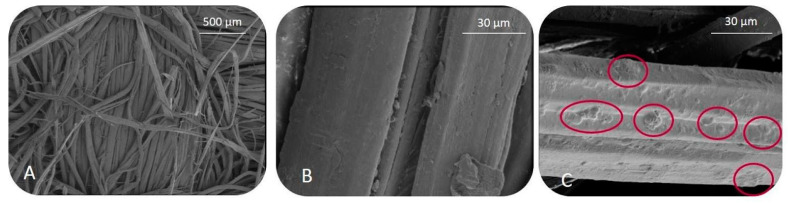
SEM images for non-impregnated (**A**,**B**) and impregnated RGPE jute fibers at (**C**). The red circles indicate the RGPE particles on the fabric surface.

**Table 1 antioxidants-10-00216-t001:** Chemical regents and microbial strains.

Reagent	Supplier
Carbon dioxide (99.99%)	Abello-Linde S.A. (Barcelona, Spain).
2,2-diphenyl-1-picrylhydrazyl (DPPH)	Sigma-Aldrich (Steinheim, Germany)
Lennox LB agar	Conda Laboratories (Torrejón de Ardoz, Spain)
Dimethyl sulfoxide (DMSO)	Panreac (Barcelona, Spain)
Phenolic standards (gallic acid, quercetin and cyanidin)	Sigma-Aldrich (Steinheim, Germany)
Lysogenic Broth (LB) with 10 g/L tryptone, 5 g/L NaCl and 5 g/L yeast extract	Sigma-Aldrich (Steinheim, Germany)
*Escherichia coli* (CECT101)	Spanish Type Culture Collection (CECT, Valencia, Spain)
*Pseudomonas aeruginosa* (ATCC 9027)	Microbiologics Inc. (Saint Cloud, MN, USA)
*Staphylococcus aureus* (ATCC 6538)	Microbiologics Inc. (Saint Cloud, MN, USA)

**Table 2 antioxidants-10-00216-t002:** Summary of the impregnation experiments’ conditions.

Experiment	P (MPa)	T (°C)	Dried RGPE (mg) *	% RGPE	% Modifier (*v*/*v*) ** (C_2_H_5_OH:H_2_O)	Molar Ratio (n CO_2_/n Modifier)	Modifier Volume (mL)
1	10	55	28.8	9.6	2.8	6.88	3
	50					20.09	
2	50	55	28.8	9.6	2.8	20.09	3
			46.1		4.8	10.73	5
			67.2		6.7	7.89	7
3	50	55	270	90	2.8	20.09	3

* Adjusted by drying RGPE using a rotavapor and then re-dissolving into the desired amount of modifier. ** Calculated respecting the total volume of the vessel.

**Table 3 antioxidants-10-00216-t003:** Antibacterial activity of pure RGPE and IJF at 50 MPa 55 °C (*n* = 2).

Bacteria	MIC RGPE (mg/mL)	Concentration of RGPE in Liquid Medium (mg/L)	% Inhibition in RGPE-Jute
*Escherichia coli*	12.0	0.07	8.250 ± 0.496
		0.14	14.877 ± 1.385
*Pseudomonas aeruginosa*	4.0	0.07	26.045 ± 3.007
		0.14	35.471 ± 1.516
*Staphylococcus aureus*	1.5	0.07	33.234 ± 1.083
		0.14	42.605 ± 1.062

**Table 4 antioxidants-10-00216-t004:** Phenolic composition of RGPE and impregnated jute fibers (*n* = 2).

	RT	Mass (Da)	RGPE (µg/mL)	IJF (µg/mL)	% Impregnation *
**Phenolic acids**					
Protocatechuic acid	0.38	153.0188	1.45 ± 0.09	5.38 ± 0.71	6.60 ± 0.47
Caffeic acid	1.81	179.0344	0.63 ± 0.90	nd	nd
*p*-coumaric acid	1.95	163.0395	1.27 ± 0.50	22.56 ± 3.46	33.08 ± 8.19
Syringic acid	1.98	197.0450	17.70 ± 1.35	22.73 ± 1.31	2.29 ± 0.31
**Flavanols**					
Catechin	2.59	289.0712	0.73 ± 0.04	20.08 ± 1.69	48.77 ± 1.63
**Flavonols**					
Rutin	1.94	609.1456	10.03 ± 3.60	nd	nd
Quercetin 3-glucoside	1.98	463.0877	43.62 ± 7.90	nd	nd
Quercetin	2.31	301.0348	20.12 ± 1.26	nd	nd
**Anthocyanins**					
Delphinidin-3-*O*-glucoside	1.98	463.0877	86.28 ± 2.04	nd	nd
Petunidin-3-*O*-glucoside	2.07	477.1033	20.99 ± 4.89	nd	nd
Delphinidin 3-*O*-(6′′-acetyl)-glucoside	2.15	505.0982	8.27 ± 0.30	nd	nd

* calculated as µg compound contained in the extract/µg compound in the fabric.

## Data Availability

All the data presented in this study are available in this article.

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
