# Peer review of "Application of a Natural Antioxidant from Grape Pomace Extract in the Development of Bioactive Jute Fibers for Food Packaging"

_antioxidants, 2021, doi:10.3390/antiox10020216_

Round 1
Reviewer 1 Report
Manuscript Revision
Title: Application of a natural antioxidant from grape pomace extract in the development of bioactive jute fibers for food packaging.
The authors of this study proposed natural jute fibers impregnated with a Petit Verdot Red Grape Pomace Extract (RGPE) as a new active food packaging material. They used high pressure techniques such as Pressurized Liquid Extraction (PLE) and Enhanced Solvent Extraction (ESE) for analyzing the extraction and impregnation process. I like the effort to the authors to elucidate Pressurized Liquid Extraction (PLE) technique offers highest RGPE bioactive extracts. From my point of view, this kind of study is really interesting because the use of materials for food packaging for the conservation of food is very important.
In the following paragraphs, I will provide clear information in order to improve the manuscript. In general, the English level is good and I cannot see many discrepancies.
Abstract
I like its contents: totally understandable, precise and concise. I could say that it is perfect and that, from my point of view, there is no discrepancy.
Introduction
The introduction is really interesting. I like the contents and the style. Only, there are few minor mistakes:
- In line 52, the words “and” and “Murr.” appear in italics in the enumeration of a plant species.
- In line 99, the word “k” should be eliminated.
Materials and Methods
Generally, this section has complete information and well explained. Here some mistakes and recommendations:
- In Table 1 (page 3) should be added their references.
- The Figure 1 (page 4) has good approach and easy understand. From my point of view, I cannot see any discrepancies.
- In line 111, add a parenthesis after the word “Spain”.
- The equal 1 (line 130), the percentage (%) should be eliminated. I consider do not necessary to add it.
- The table 2 (page 5) has complete information about the conditions of impregnation experiments. However, I advise to make any changes in that table. For example: in sixth column, I would change the word “ethanol” to “C2H5OH”” and the word “water” by “H2O”. I consider that a scientific article should have more numerical or symbolic writing.
- In line 158, I would not use so much space between N and radical.
- In equal 3 (line 164), equal 4 (line 179) and equal 7 (line 202), the percentage (%) should be eliminated.
- Improving the margin in the equals 8-10 (from line 231 to 233)
Results and Discussions
Generally, the information of this section is well explained. From my point of view, authors have performed a well-structured, clear and interesting discussion about their numerous results. Here some mistakes and recommendations have showed:
- In line 247, I would change the word “ethanol” to “C2H5OH”.
- The figure 2 (page 8) and table 3 (page 9) have excellent information and easy to understand. The table 3 (page 9) should be adjusted its position (respect the margins).
- The figure 3 (page 9) has really good visual information, but it should adjust its position (the right margin adjust more to the left).
- The table 3 (page 11) should be named as “Table 4”
- In line 412, the word “and” appears in italics in the enumeration of food pathogens.
- In line 425, the space between the words “cotton” and “natural” should be less.
- The figure 4 (page 12) has good quality images. Its position should be adjusted.
Conclusions
From my point of view, this conclusion is very well elaborated, not very long and precise. I cannot see any discrepancy.
FINAL REMARKS
In my opinion, authors have performed an extensive work, which provides interesting results of RGPE extractions from the PLE technique, with promising expectations for future research. This manuscript is clear and well written. I am suggesting MINOR REVISIONS before publishing.
Author Response
We would like to thank reviewer 1 for his comments and suggestions. These are suggestions that improve the final work. As you can see, all of them have been accepted and modified in the article.
Corrections Reviewer #1
- In line 52, the words “and” and “Murr.” appear in italics in the enumeration of a plant species.
Amended.
- In line 99, the word “k” should be eliminated.
Amended.
- In Table 1 (page 3) should be added their references.
Now Table 1 contains the reagents and the commercial brands.
- The Figure 1 (page 4) has good approach and easy understand. From my point of view, I cannot see any discrepancies.
Authors thank the reviewer comments.
- In line 111, add a parenthesis after the word “Spain”.
Amended.
- The equal 1 (line 130), the percentage (%) should be eliminated. I consider do not necessary to add it.
Amended.
- The table 2 (page 5) has complete information about the conditions of impregnation experiments. However, I advise to make any changes in that table. For example: in sixth column, I would change the word “ethanol” to “C2H5OH”” and the word “water” by “H2O”. I consider that a scientific article should have more numerical or symbolic writing.
Amended. Authors agree with the reviewer and the word “ethanol” and “water” have been modified by their formula along the manuscript when required. (table 2, line 97, line 129 line 247, line 268, line 284, line 288, line 302, line 314, line 324, line 342, line 371, line 387, line 390 and line 435 as well as in the extraction conditions showed in figure 2 and the abstract.)
- In line 158, I would not use so much space between N and radical.
Amended.
- In equal 3 (line 164), equal 4 (line 179) and equal 7 (line 202), the percentage (%) should be eliminated.
Amended.
- Improving the margin in the equals 8-10 (from line 231 to 233)
Amended.
- In line 247, I would change the word “ethanol” to “C2H5OH”.
Amended.
- The figure 2 (page 8) and table 3 (page 9) have excellent information and easy to understand.
Authors thank the reviewer comments.
- The table 3 (page 9) should be adjusted its position (respect the margins).
Amended.
- The figure 3 (page 9) has really good visual information, but it should adjust its position (the right margin adjust more to the left).
Amended.
- The table 3 (page 11) should be named as “Table 4”
Amended.
- In line 412, the word “and” appears in italics in the enumeration of food pathogens.
Amended.
- In line 425, the space between the words “cotton” and “natural” should be less.
Amended.
- The figure 4 (page 12) has good quality images. Its position should be adjusted.
Amended
Reviewer 2 Report
-This paper is focused on a natural antioxidant from grape pomace extract in the development of bioactive jute fibers for food packaging. It is an acceptable paper, particularly the agricultural processing research field of reuse of food byproducts.
-The research content is innovative and novel, and the research structure is complete.
-The only minor problems are that 2.6. Scanning Electron Microscopy (SEM) samples are not mentioned in the front preparation and 2.7. Statistical Analysis is too rough.
Author Response
- This paper is focused on a natural antioxidant from grape pomace extract in the development of bioactive jute fibers for food packaging. It is an acceptable paper, particularly the agricultural processing research field of reuse of food byproducts.
The research content is innovative and novel, and the research structure is complete.
Authors thank the reviewer comments.
- The only minor problems are that 2.6. Scanning Electron Microscopy (SEM) samples are not mentioned in the front preparation and 2.7. Statistical Analysis is too rough.
Amended. The sentence “Samples analyzed were those obtained at the most convenient impregnation conditions” was added in the section 2.6.
Correction. Authors agree with the reviewer comments. The statistical analysis is dispensable considering the data obtained, since the differences among the samples are perceived attending to the deviation of the measurements. Thus, the ANOVA analysis did not provide complementary information in either figure 2 or 3. Therefore, the section 2.7. has been removed.